# MicroRNA in the Diagnosis and Treatment of Doxorubicin-Induced Cardiotoxicity

**DOI:** 10.3390/biom13030568

**Published:** 2023-03-20

**Authors:** Ziyu Kuang, Jingyuan Wu, Ying Tan, Guanghui Zhu, Jie Li, Min Wu

**Affiliations:** 1Oncology Department, Guang’anmen Hospital, China Academy of Chinese Medical Sciences, Beijing 100053, China; 2Graduate School, Beijing University of Chinese Medicine, Beijing 100029, China; 3Cardiovascular Department, Guang’anmen Hospital, China Academy of Chinese Medical Sciences, Beijing 100053, China

**Keywords:** doxorubicin-induced cardiotoxicity, microRNA, biomarker, therapeutic target

## Abstract

Doxorubicin (DOX), a broad-spectrum chemotherapy drug, is widely applied to the treatment of cancer; however, DOX-induced cardiotoxicity (DIC) limits its clinical therapeutic utility. However, it is difficult to monitor and detect DIC at an early stage using conventional detection methods. Thus, sensitive, accurate, and specific methods of diagnosis and treatment are important in clinical practice. MicroRNAs (miRNAs) belong to non-coding RNAs (ncRNAs) and are stable and easy to detect. Moreover, miRNAs are expected to become biomarkers and therapeutic targets for DIC; thus, there are currently many studies focusing on the role of miRNAs in DIC. In this review, we list the prominent studies on the diagnosis and treatment of miRNAs in DIC, explore the feasibility and difficulties of using miRNAs as diagnostic biomarkers and therapeutic targets, and provide recommendations for future research.

## 1. Introduction

DOX, a commonly used anthracycline, is widely applied to the clinical treatment of breast cancer, gastric cancer, lymphoma, and leukemia, etc. [1]. However, tumor treatment-related cardiovascular side effects are not to be ignored [2], with DIC being the most common [3]. DIC was first reported in 1967 and is displayed as arrhythmia, coronary artery disease (CAD), cardiomegaly (CM), hypertension, and thromboembolism [4]. These cardiovascular side effects can be long-lasting and fatal [5] and have the characteristics of high morbidity and irreversibility [6]. DOX remains the therapeutic cornerstone for many malignant tumors [7,8]. Therefore, early monitoring and diagnosis of DIC are essential. A diagnosis of DIC is made by examining the patient’s scans, serum biomarkers, medication conditions, and signs and symptoms; however, these detection techniques have limitations, including low specificity, high detection cost, and late detection of cardiac dysfunction [9,10]. Therefore, stable, highly specific, and highly sensitive biomarkers for assisted diagnoses are required. Patients with cardiotoxicity or risk of cardiotoxicity are usually treated with cardioprotective drugs, which provide only symptomatic treatment [11]. Currently, researchers are actively exploring new therapeutic methods to further enrich the treatment of DIC.

Indeed, miRNAs possess many biological functions and participate in many biological processes of the heart, including differentiation, replication, and regeneration [12,13]. With the development of gene sequencing technology and the emphasis on miRNA research, miRNAs are expected to act as specific biomarkers and therapeutic targets for DIC [14]. The role of miRNAs in the early prediction and diagnosis of DIC could provide the possibility for the application of therapeutic targets in DIC shortly. In this review, the latest research progress on miRNAs in DOX-induced cardiotoxicity is detailed to illustrate the potential of miRNAs as DIC biomarkers and therapeutic targets.

## 2. miRNAs: Regulatory Factors of the Heart

### 2.1. Structure and Function of miRNAs

miRNAs consist of approximately 19 to 25 nucleotides, are usually only 21 to 23 base pairs in length, and are evolutionarily conservative, non-coding single-stranded RNA [15]. The first miRNA, lin-4, was discovered in worms in 1993, and the presence of miRNAs was detected in human plasma and serum in 2008 [16]. After continuous research and exploration, miRNAs have been gradually isolated from cells, tissues, and bodily fluids (including urine, saliva, and cerebrospinal fluid). Currently, more than 2000 kinds of miRNAs have been discovered [17].

The synthesis and processing maturation of miRNAs requires the coordination of multiple enzymes and proteins. The transcription of miRNAs is mediated primarily by RNA polymerase II, while a few are mediated by RNA polymerase III. Initially, RNA polymerase synthesizes primary miRNAs (pri-miRNAs) with miRNAs, which usually consist of thousands of nucleotides and contain an internal stem-loop [18]. Two important “cuts” are required for the pri-miRNA to mature to miRNA. First, the stem-loop on the pri-miRNA is “cut” from the initial transcript by double-stranded RNase III (DROSHA) in the nucleus, yielding a hairpin precursor miRNA (pre-miRNA), which is double-stranded with a length of about 70 nucleotides [19]. Subsequently, the pre-miRNA is transported by exportin-5 into the cytoplasm and cleaved by the endonuclease DICER (an enzyme of RNase III) into a double-stranded RNA of approximately 22 nucleotides [20]. Helicase further converts the double-stranded miRNA into single-stranded miRNA. Then, the single-stranded miRNA forms a multi-protein RNA-induced silencing complex (RISC) together with the Argonaute (AGO) protein [21]. Subsequently, the more unstable strand of the double strand is often retained, which represents the mature miRNA, while the other strand is degraded (Figure 1). 

Moreover, miRNAs are closely related to gene expression [22]. For example, following the formation of the RISC complex, miRNAs bind to a specific mRNA, which causes the mRNA to break, and blocks the mRNA translation into protein [23]. The role of miRNAs is bidirectional, whereby a miRNA can simultaneously target multiple genes located in the signaling pathway of the same cell, while multiple miRNAs can also coordinately affect a single mRNA. This interaction depends on the location of the miRNA, the expression of miRNA and target mRNA, and the compatibility of the interaction [24]. At least 30% of the human protein-encoded genome is regulated by miRNAs [25]. Additionally, miRNAs can affect a series of biological processes in the human body, such as cell proliferation, differentiation, and apoptosis [26].

### 2.2. Characteristics of miRNAs

The imbalance of miRNAs is crucial in the occurrence and development of cancer, cardiovascular disease (CVD), and metabolic diseases [27]. For example, non-small cell lung cancer cell lines secrete extracellular vesicles (EVs) that contain miR-21/29a, and these enriched EVs are associated with the activation of the tumor nuclear factor κB (NF-κB) and the release of the pre-metastatic inflammatory cytokines, tumor necrosis factor α (TNF-α) and interleukin 6 (IL-6) [28]. Patients with familial hypercholesterolemia exhibit upregulated miR-223/105/106a in their high-density lipoprotein (HDL) particles. In cultured hepatocytes, HDL-targeted receptor scavenger receptor class B member 1 (SRB1) mRNA reduces intrahepatic lipoprotein cholesterol [29]. In addition, studies have detected tissue-specific expression of miR-122 in the liver, miR-9 and miR-124 in the brain, and miR-192-5p in the colon. Tissue-specific miRNA is usually associated with specific tissue-related diseases [30].

The miRNAs are mainly secreted into the extracellular environment in exosomes or in vesicles for long-distance intercellular communications [31]. Circulating miRNAs are stored in small membranous vesicles such as exosomes [32], exosome-like vesicles [33], and apoptotic bodies [32], which are surrounded by cholesterol or bound to RNA-binding proteins. Even under some extreme conditions, such as repeated freeze–thawing, high temperatures, PH changes, and long-term storage, their structures are not damaged, showing strong anti-degradation performance [34]. In vivo miRNAs can be accurately determined using techniques such as real-time quantitative polymerase chain reaction (RT-qPCR). In addition, some toxic substances can cause changes in miRNAs in heart tissue and plasma, and changes in miRNA can even be measured at low concentrations of the toxin, without significant changes in other biomarkers [30]. Given the broad distribution, ease of detection, high sensitivity, and high stability of miRNA, its use is extremely suitable for the early diagnosis and prognosis prediction of known diseases and syndromes.

### 2.3. miRNAs in Cardiovascular Physiology and Pathology

Although there are many kinds of miRNAs, specific miRNAs account for most of the miRNAs in the cardiovascular. These specific miRNAs are either abundant and participate in maintaining the function of the cardiovascular, or they are limited in number, but they play an important role in the regulation of the cardiovascular [35]. miR-1 and miR-133, both belonging to a cluster, are highly enriched in the early stage of cardiac development, mainly controlling the early stage of cardiogenesis, synergistically promoting the differentiation of embryonic stem cells into mesodermal cells, and maintaining myocardial electrophysiology [36]. miR-1 is the most abundant miRNA in the heart, also highly abundant in myocardial cells in the left atrium (LA) and ventricle (LV) [37], which is a negative regulator of proliferation and participates in the regulation of heart cell growth and development. For example, overexpressed miR-1 inhibits cyclin-dependent kinase (CDK)6, resulting in mouse Purkinje fibrous dysplasia [38]. At the same time, overexpressed miR-1 can also downregulate the downstream target Hes1 of the Notch signaling pathway and promote the differentiation of mesenchymal stem cells into heart cell lineage [39]. Oncostatin M receptor (Osmr) and Fibroblast growth factor receptor (Fgfr1) are important in regulating the dedifferentiation and proliferation of cardiomyocytes. miR-1 often cooperates with miR-133 to inhibit the expression of Osmr and Fgfr1 after transcription, thus being the key regulatory factors of dedifferentiation and cell cycle entry of cardiomyocytes, inhibiting dedifferentiation, and cell cycle [40]. miR-133 is also one of the miRNAs with the most myocardial cells. Similar to miR-1, miR-133 is also a negative regulator of cardiac proliferation. miR-133 is involved in many pathological processes of the heart, such as mediating cardiac hypertrophy [41]. Snai1 is the main regulator of epithelial-mesenchymal transition. Over-expression of miR-133 upregulates the expression of Snai1 and mediates fibroblasts to complete the reprogramming process to cardiomyocyte-like cells [42]. Another meta-analysis showed that miR-133 was associated with the inhibition of cardiac hypertrophy [43]. miR-1 and miR-133 are two of the most important miRNAs in the heart; they work alone or in combination to regulate the development and differentiation of myocardial cells and maintain the normal function of the heart. miR-208a, miR-208b, and miR-499 are encoded by the myosin gene [44] and are also highly enriched in the cardiovascular system, although at lower levels than miR-1 and miR-133. miR-208a-3p and miR-208a-5p are abundant in the LA, while miR-208b-3p and miR-208b-5p are expressed earlier in the LV [37]. The miR-208 family is encoded by the α-myosin heavy chain (MHC) gene and participates in the regulation of the balance between α-MHC and β-MHC [45]. miR-208 is a cardioprotective miRNA that targets PDCD4, downregulates Bcl-2 associated-x (Bax) and cleaves caspase-3, upregulates B cell lymphoma-2(Bcl-2) and caspase-3, and inhibits cardiomyocyte apoptosis [46]. miR-499 promotes the differentiation of cardiomyocyte progenitor cells, regulates the proliferation and apoptosis in the intermediate and late stages of cardiac differentiation by regulating Sox19 and cyclin D1 [47], and inhibits the expression of SOX6 to promote the proliferation and migration of vascular smooth muscle cells (VSMCs) [48]. miR-1, miR-133, miR-208, and miR-499 are heart-related miRNAs, also known as myomiRs [49], playing an important role in the physiological and pathological regulation of the heart.

Some miRNAs are less abundant in the heart, but they are essential for cardiac homeostasis. As a cardioprotective factor, miR-21 is especially abundant in fibroblasts [50], inhibiting cardiomyocyte apoptosis by upregulating Bcl-2 and CDK6 expressions [51]. In addition, studies have also shown that miR-21 is the miRNA with the highest expression level in cardiac macrophages (cMPs), and miR-21 in cMPs is crucial for their polarization to the M1-like phenotype [52]. Targeting delivery of therapeutic miR-21 has been reported to improve the recovery of cardiac function after myocardial infarction [53]. However, miR-21 is also overexpressed in a variety of diseases, including cancer, so it is also regarded as an oncomiR [54]. miR-34, as an aging-related miRNA, is related to cardiac dysfunction and related cardiovascular diseases. miR-34a targets Smad4/TGF-β1 to promote cardiac fibrosis [55], and upregulation of SIRT1 mediates atherosclerosis as well as myocardial cell senescence caused by chemotherapy [56,57]. Notably, miR-34 does not seem to be a heart-specific miRNA, and the miR-34 family is also widely involved in the regulation of various diseases, such as cancer [58] and Alzheimer’s disease [59]. In addition, many heart-related miRNAs, including lethal-7(Let-7) [60], miR-15 cluster [61], miR-17-92 cluster [62], miR-140 [63], and miR-145 [64], are also involved in the regulation of heart function, but they still need to be further explored.

## 3. DIC: Mechanisms, Diagnosis, and Treatment

### 3.1. Mechanisms of DIC

The relatively clear mechanisms studied at this stage include oxidative stress caused by DOX and the roles of DOX and topoisomerase [65]. As the current research expands, mechanisms such as epigenetic modification, cardiac progenitor cells (CPCs), and endoplasmic reticulum stress (ERSR) are continually being discovered and studied; however, the mechanism of DIC has not yet been fully revealed and requires further research (Figure 2).

#### 3.1.1. Oxidative Stress

Oxidative stress plays an important role in the occurrence and development of DIC and is considered to be one of the main mechanisms for DIC. The central mechanism of oxidative stress, which has been induced by DOX, is the imbalance between reactive oxygen species (ROS) and antioxidant enzymes in cardiac cells [66]. Simply, DOX is involved in the stress responses mainly through the following pathways: (1) The nitric oxide synthase (NOS)-dependent increase in ROS production, where DOX combines with endothelial nitric oxide synthase reductase (eNOS) to induce DOX, generating semiquinone free radicals and reducing the free oxygen to superoxide radicals (O^2−^) [67]. (2) Mitochondrial-dependent ROS production: mitochondria are the main sites of ROS production. DOX has a strong affinity with the mitochondrial inner membrane and can form irreversible complexes, which change the function of cardiolipin at the interface and lead to the formation of O^2−^ [68]. (3) Fe–DOX complex: in populations with iron regulatory protein (IRP-1) gene deficiency, Fe levels are elevated, and DOX forms a Fe–DOX complex with Fe, which contributes to the further generation of hydroxyl radicals. Hydroxyl radicals destroy DNA and protein, and extensively oxidize lipids, leading to major cell injury and death [69]. (4) NADPH-dependent ROS: DOX reacts with NADPH to generate semiquinone free radicals [70], under the action of the NADPH oxidase (NOX) enzyme. (5) Intracellular calcium dysregulation: during the metabolism of DOX, the toxic metabolite DOXOL is formed, which inhibits the sodium–calcium exchange channel, leading to an imbalance of calcium levels, an increase of intracellular calcium, and the production of ROS. In addition, CalcIUM/calmodulin-dependent protein kinase II (CaMKII) is a major Ca^2+^ regulatory protein that can be oxidized and activated by free radicals, thus, further leading to an intracellular Ca^2+^ imbalance [71]. This action can change the permeability of the cell’s membrane and mitochondria. (6) The loss of antioxidants: DOX significantly reduced the levels of endogenous antioxidants, glutathione, and catalase, resulting in redox imbalance and increased oxidative stress [72]. (7) Downregulation of the Nrf2 pathway: DOX downregulated Nrf2 and aggravated oxidative stress [73]. (8) Inhibitory effect of DOX on acetylation: The high expression of the silent information regulator 1 (SIRT1) could restore DOX-induced deubiquitination of p53 and reduce the activation of caspase-3 while downregulating ROS production [74]. However, DOX treatment significantly inhibited SIRT1 deacetylase activity and protein levels.

Oxidative stress is crucial in the occurrence and development of DIC. DOX acts on cardiomyocytes or vascular endothelial cells and mediates the occurrence of the oxidative stress response in heart tissue through the aforementioned pathways, either alone or in combination, to cause direct or indirect mitochondrial damage [75], DNA damage [76], inflammatory responses [77], lipid peroxidation [78], ferroptosis [79], Ca^2+^ homeostasis destruction [80], cell apoptosis [81], autophagy [82], apoptosis [83], etc. (Figure 3).

#### 3.1.2. The Role of Topoisomerase in DIC

Topoisomerases are usually divided into two types [84], topoisomerase I (topo I) and topoisomerase II (topo II), which are responsible for altering the structure of DNA single strands and DNA double strands, respectively. Topo IIα was overexpressed only in tumor cells, whilst not detected in cardiomyocytes. The high expression of topoisomerase IIα in tumor cells may be related to the efficacy of DOX as a chemotherapeutic agent. In contrast, adult cardiomyocytes mainly express topo IIβ, while topo IIβ is also expressed in myocardial tissue [85]; thus, the occurrence of DIC is mainly related to topo IIβ [86].

The effect of DOX on cells is non-selective. Further, DOX embeds DNA base pairs, resulting in the rupture of double-stranded DNA, the initiation of a DNA damage reaction, and programmed cell death [87]. In addition, DOX, working as a topoisomerase inhibitor, can inhibit DNA replication and transcription by inhibiting topoisomerase [88], thus, promoting tumor cell death. Conversely, it may also damage the structure and function of normal heart cells, which can lead to DNA structure damage in cardiomyocytes and cardiovascular intima cells. DOX forms a topo IIβ-DOX–DNA complex with topo IIβ and DNA chains, which induces DNA double-strand breaks in cardiomyocytes and triggers apoptosis [89]. Patients who exhibit higher levels of topo IIβ are more likely to develop cardiotoxicity during DOX chemotherapy. 

#### 3.1.3. Other Potential Mechanisms

DNA methylation is a pre-transcriptional heritable modification that causes changes in chromatin structure, DNA conformation, DNA stability, and the mode of interaction between DNA and protein, thereby controlling gene expression [90]. DOX downregulates DNA methyltransferase 1 (DNMT1) enzyme activity, which decreases the DNA methylation process [91]. Furthermore, DOX reduces DNA methylation in heart tissue by affecting mitochondrial function.

Although the heart is considered to be a terminally differentiated organ, it still has a certain capacity to repair after injury, which is achieved through CPCs. DOX can cause the apoptosis of CPCS and inhibit their differentiation into cardiomyocytes, while also reducing the cardiac repair of endothelial cells [92,93]. The injury of CPCs is considered an additional potential mechanism of DIC.

The endoplasmic reticulum (ER) plays an important role in protein folding and calcium homeostasis [94,95], while DOX-induced ER dysfunction leads to the accumulation of unfolded proteins and calcium interference [96]. ERSR regulates DOX by inducing autophagy and apoptosis through the IRE1α/JNK/beclin-1 signaling pathway, and the IRE1α/ASK1/JNK and CHOP signaling pathways, respectively. Therefore, ERSR is one of the main factors in cardiac complications resulting from DOX [97].

### 3.2. Diagnosis and Treatment of DIC

Clinically, DIC may be acute, early, and chronic. Acute DIC occurs within minutes to a week after a single dose or course of treatment, and early DIC usually occurs within the first year of DOX treatment, while chronic cardiotoxicity manifests after several years of treatment (median of seven years) [98]. The ESMO consensus recommendation issued in 2020 clearly pointed out that asymptomatic patients who had been treated with cardiotoxic drugs and had normal cardiac function should consider screening for new asymptomatic LV dysfunction 6–12 months after treatment and 2 years after treatment, and carry out potential cardiac imaging examination or screening regularly after treatment [99]. At present, cardiac function evaluation includes serum myocardial markers (troponin, BNP), electrocardiogram (long-range electrocardiogram, derived calculation formula [100]), and echocardiography. However, the diagnostic methods of DIC also have limitations. Because of the early compensation of the heart after injury, imaging examination cannot reflect cardiac toxicity in time [9,10]. Although there are clinical studies showing that plasma NP levels, especially NT-pro BNP, can be used as biomarkers of cardiotoxicity induced by acute chemotherapy [9,10], cTnT and cTnI can be used as a complement to NP; they belong to paraclinical markers released by cardiovascular system. 

Currently, treatments for DIC include alterations in the chemotherapy regimen, liposomal formulations using DOX, mineralocorticoid receptor antagonists, standard heart failure drugs (angiotensin-converting enzyme inhibitors, angiotensin receptor blockers, and beta blockers), and dexrazoxane (DEX). DEX is the only FDA-approved drug for the treatment of DIC, and after intracellular hydrolysis, dexrazoxane chelates with intracellular iron reduce trivalent iron ions to form complexes with anthracyclines, and prevent free radical formation, thus playing a protective role [101]. Inhibition of cardiac topo IIβ and inhibition of DOX-induced DNA fragmentation are also possible mechanisms of dexrazoxane cardioprotection [102]. However, dexrazoxane may be associated with reduced anti-tumor efficacy and increased risk of secondary malignancies [103]. The FDA has approved usage of dexrazoxane restricted to cardioprotection in advanced or metastatic breast cancer with ongoing anthracycline use after a cumulative dose of greater than 300 mg/m^2^ of DOX. However, these agents are unlikely to address specific molecular pathways involved in DIC. Therefore, exploring alternative pharmacological treatments for DIC is urgently needed.

## 4. miRNA as a Biomarker of DIC

### 4.1. miR-1

miR-1 is considered to be a muscle-specific miRNA and is the most specific and richly expressed miRNA in heart tissue [104,105,106]. Studies have indicated that the serum level of miR-1 in patients with acute MI and unstable angina pectoris is significantly increased, even without the increase of serum creatine phosphokinase (CK), cardiac troponin (cTn), or other biomarkers [107].

An animal study [108] divided the tumor mice into control, liposome, DOX (6 and 9 mg/kg), and pegylated liposomal DOX (PL-DOX) (6 and 9 mg/kg). RT-qPCR showed a significant 2.14-fold increase in DOX (9 mg/kg) miR-1 expression, upregulating the expression of Bax, caspase-3, and caspase-8, and suppressing Bcl-2 expression after dosing in the fourth week. Bcl-2 expression was not significantly changed after PL-DOX injection compared with the control group. Bcl-2 is considered to be a downstream target of miR-1, indicating that miR-1 may play a role in DOX-induced cardiac apoptosis by targeting Bcl-2. A study by Nishimura et al. [109] treated rats with DOX (30 mg/kg) and collected blood samples after 2, 4, 8, and 24 h. Studies showed that miR-1 levels in the venous blood of rats treated with DOX for 8 h were elevated (*p <* 0.01). Another study [110] treated in vitro human pluripotent stem cells (hPSC)-ventricular cardiomyocytes with DOX (1 μmol/L) for 48h, which resulted in a dose and time-dependent upregulation of miR-1.

miR-1 provides a good early diagnosis, while compared with other commonly used biomarkers, it can also be used as a prognostic factor for DIC. A prospective cohort study [111] showed that 15 of the 24 patients that demonstrated cardiotoxicity after their anthracycline treatment had used DOX. miR-1 in patients’ plasma was detected at three time points (6, 12, and 24 h after treatment). Results showed that plasma miR-1 was upregulated at all three time points, irrespective of an increase in plasma cardiac troponin T (cTnT) levels when compared to patients without cardiotoxicity. In another study [112], 10 of the 56 breast cancer patients treated with DOX chemotherapy experienced cardiotoxicity, and LVEF decreased from 67.2 ± 1.0 (baseline) to 58.8 ± 2.7 (12 months; *p* = 0.005), although there were no differences in the plasma cardiac troponin I (cTnI) levels between the cardiotoxic and non-cardiotoxic patients. Comparison of these patients with and without cardiotoxicity illustrated that only miR-1 showed a differential pattern after cycle 2, while its expression was also significantly increased in patients with cardiotoxicity. A receiver operating characteristic (ROC) curve analysis, comparing the area under the curve (AUC) of miR-1 (AUC = 0.8510), illustrated that it was greater than that of cTnI (AUC = 0.544) (*p* = 0.0016), and that miR-1 was associated with left ventricular ejection fraction (LVEF) changes (*p <* 0.001). Even after administering a safe dose of DOX treatment, cardiac toxicity still existed in 17.9% of patients, while using miR-1 as a biomarker of DIC did illustrate DIC with a higher sensitivity than cTnI. This study first confirmed that miR-1 was related to LVEF changes, indicating that miR-1 was related to the cardiac function prognosis of DIC. A study by Cheung et al. [110] followed 39 lymphoma patients for up to 6 months. Here, the global longitudinal strain (GLS) decreased significantly within 1 week after DOX (*p <* 0.05) when the cTnT level peaked. Moreover, miR-1 significantly doubled over the treatment period relative to the baseline, with plasma miR-1 increasing an average of 4.6-fold over a 24-h period of dosing (*p* = 0.055), 4.2-fold over 1 week after the first chemotherapy (*p* = 0.006), and 7.7-fold over 6 months after chemotherapy (*p* = 0.008). In addition, there is evidence-based medicine that demonstrates that miR-1 is differentially expressed in the DIC and non-cardiotoxic groups, although the changes in its levels are controversial [113].

### 4.2. Let-7

Let-7 is involved in the development and differentiation of embryonic stem cells in the cardiovascular system, and the occurrence and development of CVD [114,115]. Let-7a and let-7e downregulate the expression of β1 adrenergic receptors in the cardiac myocytes of rats with HF, while also regulating its downstream signaling pathway [116]. Inhibiting the expression of let-7a and-7e can significantly increase cardiac fibrosis and HF [117]. At present, most of the research on let-7 focuses on animal and cell experiments to expound the expression of the let-7 family in different concentrations and different parts of the DIC model. Many studies show that let-7 is downregulated in the DIC model.

Fu et al. [118] divided the study into two parts. The in vivo experiment indicated that the heart rate of rats in all DOX-treated with 24 h groups decreased, pulse pressure increased (*p <* 0.05), the concentration of cTnT was significantly increased, and the level of let-7 was decreased (*p <* 0.05). Additionally, data from in vitro studies illustrate that the content of let-7g was downregulated in all DOX-treated cells (*p <* 0.05). Chen et al. [119] explored the in vitro changes of the let-7 family in DIC by exposing H9c2 cardiomyocytes to DOX (5 µg/mL) for 0, 12, and 24 h, and showed that the expression levels of the seven members of let-7 were downregulated. Another study [120] divided male Wistar rats into control, DOX (5 mg/kg), and liposome DOX (L-DOX) (5 mg/kg) groups and sacrificed these 24 h later to collect the left atrium (LA) and left ventricle (LV) alongside the right atrium (RA) and right ventricle (RV) samples. In the control group, let-7g was expressed only in the atrial–ventricular gradient, and mainly in the atrium. Both DOX and L-DOX resulted in a 13.79% and 14.70% (*p* = 0.015) decrease in the expression of let-7g in the LA, respectively. To further explore the role of let-7a in DIC, A431 cells and HepG2 cells were treated with DOX (0.05 μg/mL and 0.2 μg/mL, respectively), and both cell lines were transfected with let-7a. Luciferase report results showed that let-7a binds to the 3’-UTR of caspase-3, inhibiting caspase-3 expression, thereby reducing DOX-mediated apoptosis of cardiomyocytes [121]. In addition, a meta-analysis showed that let-7 was downregulated in the circulation of patients [113].

### 4.3. miR-21

miR-21 has attracted attention due to its various effects on cardiac function [122,123]. Some studies showed that miR-21 had a protective effect on ischemia-induced cell apoptosis that was associated with its target gene programmed cell death 4 (PDCD4) [124]. Some studies revealed that miR-21 acts as a protective factor for the heart and protects the cardiomyocytes from DOX-induced toxicity.

Studies have shown that miR-21 exerts a protective effect against ischemia-induced apoptosis by binding to its target genes’ programmed cell death protein 4 (PDCD4) and activating protein-1 (AP-1) [125]. In an animal study [108], tumorous mice were divided into the control group, liposome group, DOX (6 and 9 mg/kg) group, and PL-DOX (6 and 9 mg/kg) group: the protein expression of PDCD4 was inhibited in the DOX (9 mg/kg) group compared with the control group (*p <* 0.05), and there was no change in PDCD4 protein expression in PL-DOX-exposed animals. RT-qPCR demonstrated an upregulation of miR-21 levels in the DOX (9 mg/kg) group in the fourth week (*p <* 0.05). Tong et al. [126] divided the mice into four groups: acute DOX injury group (A-DOX), acute normal saline control group (A-NS), chronic DOX injury group (C-DOX), and chronic normal saline control group (C-NS), and detected the expression of miR-21 and B cell translocation gene 2 (BTG2) in DOX-treated cardiomyocytes. In the C-DOX group, the expression of miR-21 was 1.8 times higher than in the C-NS group (*p <* 0.05); however, there was no statistical difference in expression levels of miR-21 between the A-DOX group and the A-NS group (*p* > 0.05). Compared with A-DOX group, the number of apoptotic cells in C-DOX group increased significantly. Moreover, BTG2 expression was significantly reduced in both the A-DOX group (*p <* 0.01) and the C-DOX group (*p <* 0.01). These results indicate that BTG2 is a target gene of miR-21 in cardiomyocytes and that miR-21 may protect cardiomyocytes from DOX-induced injury through the post-transcriptional regulation of BTG2.

### 4.4. miR-34

The overexpression of the miR-34 family, directly and indirectly, aggravates the process of vascular aging [56], which is closely related to cardiovascular fibrosis [127], and MI [128]. Furthermore, the role of the miR-34 family has been widely recognized in DIC [129]. The diagnostic potential of miR-34 for DIC is illustrated through both in vivo and in vitro experiments, and by establishing early-onset and late-onset DIC models that describe the expression of the miR-34 family in DIC and compare the sensitivity of miR-34 with that of traditional detection methods.

An in vivo study [130] divided mice into a DOX group (3 mg/kg for 24h) and a control group and analyzed the subsequent plasma samples. These revealed increased cTnT levels and a significantly upregulated expression of miR-34a-5p in the DOX group compared with the control group (*p <* 0.001). Similarly, Gioffré et al. [131] divided the experiment into two sections, whereby in vivo, mice were given normal saline or DOX for 14d, while an in vitro study analyzed the miRNA in cell culture medium after murine immortalized cardiomyocytes (HL-1) were treated with DOX (1 μmol/L for 24 h). The in vivo study showed a marked decrease in LVEF (*p* = 0.01) and left ventricular fractional shortening (LVFS) (*p* = 0.03) in DOX-treated mice and an upregulation of miR-34a-5p in DOX-treated mice atria at the end of treatment. Comparatively, the in vitro study showed that the expressions of miR-34a-5p, miR-34b-3p, and miR-34c-5p were all upregulated. Desai et al. [132] explored the expression of miR-34a-5p by modeling early-onset and late-onset cardiotoxicity using DOX. After one week of treatment with DOX in the early-onset model, only the expression of miR-34a-5p was increased (false discovery rate (FDR) < 0.1), and there were statistically significant dose-related reactions with all the total cumulative doses. In the delayed cardiotoxicity model, miR-34a-5p was expressed more in the plasma and heart than in the normal saline group (FDR < 0.1), suggesting that miR-34a-5p had the potential to be an early biomarker of delayed onset cardiotoxicity. Holmgren et al. [133] exposed cardiomyocytes to DOX at various concentrations (50, 150, and 450 nmol/L) and collected cells for miRNA and protein analysis after exposure initiation, miR-34a and miR-34b were continuously expressed at four time points (days 1, 2, 7, and 14) under different doses of DOX. In another study [134], mice were divided into the 3 mg/kg DOX group (n = 12) and the normal saline group (n = 10). The results showed cTnT was increased and myocardial damage was confirmed (cumulative DOX doses of 18 mg/kg and higher), while myocardial lesions were observed under a light microscope in all mice (cumulative DOX dose of 24 mg/kg). miR-34a expression increased at the lowest cumulative dose of DOX (6 mg/kg, 3 mg/kg/week, for 2 weeks), 2.2-fold following a cumulative dose of 18 mg/kg, and 2.9-fold after a cumulative dose of 24 mg/kg, whereas plasma cTnT concentrations were only significantly increased following a cumulative dose of 18 mg/kg. This study shows that an increase of miR-34a occurs before the increase of cTnT. Some studies have attempted to explore the mechanism by which the miR-34 family induces DIC. After treatment of HL-I cells with different concentrations of DOX (0.01, 0.1, 1, and 10 μM/L), their cell viability decreased with increasing DOX concentration and miR-34b/c expression gradually increased. As the dose of DOX increased, the mRNA expression of ITCH was gradually downregulated, the mRNA expression of NF-κB protein and TNF-α and IL-6 was upregulated, and silencing ITCH promoted protein levels of total NF-κB, TNF-α, and IL-6. The expression of miR-34b/c was negatively correlated with the expression of mRNA of ITCH, and the double luciferase reporter showed that ITCH was a downstream target of miR-34b/c, and miR-34b/c could increase the expression of NF-κB, TNF-α and IL-6 by silencing ITCH [135]. Under DOX treatment of 1.0–5.0 μM/L, the degree of apoptosis of H2c9 cells was positively correlated with the dose, and miR-34a-5p was overexpressed; further research found that overexpression of miR-34a-5p could significantly increase the expression of Bax mRNA and significantly increase mitochondrial depolarization, while diluciferase experiments showed that miR-34a-5p significantly downregulated its expression by binding to SIRT1 3’-UTR; P66shc is the target gene of SIRT1. P66shc mediates the mitochondrial cell death pathway by increasing lipid peroxidation-induced apoptosis; transfection of miR-34a-5p mimic can effectively inhibit the expression of SIRT1 protein and Bcl-2, enhance p66shc expression, caspase-3 expression, and Bax and activation [136].

A study of breast cancer patients [137] showed that miR-34a-5p exhibited a DOX-induced significant upregulation at each treatment time point, regardless of cTn. Notably, an increase in cTn after administration may indicate an increased propensity to develop cardiac dysfunction; however, it does not provide any information on the timing at which it may occur [138].

### 4.5. miR-133

Similar to miR-1, miR-133 is also a heart-enriched miRNA involved in myocardial reprogramming and physiopathological processes in the heart [139], and miR-133a has been shown in meta-analyses to be elevated in patients with non-ST-segment elevation myocardial infarction and may be a potential biomarker [140].

The apoptosis rate of HL-1 cardiomyocytes increased significantly after treatment with DOX (2.5, 5, and 10 μM) for 6h, and miR-133b expression levels were significantly reduced according to RT-qPCR results. A mouse model of heart injury was then established by injecting DOX. Echocardiographic data showed that after doxorubicin injection, left ventricular end-diastolic dimension (LVEDD) and left ventricular end-systolic diameter (LVESD) were significantly increased, while LVEF and LVES were significantly reduced after DOX injection. In addition, the expression of miR-133b in the DOX group was significantly downregulated compared with the control group. Further exploring the mechanism, western blot detection showed that miR-133b overexpression upregulated the expression of Bcl-2 protein and downregulated the expression of Bax and cleaved caspase-3 protein. PTBP1 and TAGLN2 are targets of miR-133b, which exacerbate cardiomyocyte apoptosis by increasing the expression levels of the apoptotic proteins caspase-8, caspase-9, and caspase 3 and decreasing the expression of Bcl-2. miR-133b binds to the 3’-UTR of PTBP1 and TAGLN2 mRNAs to regulate downstream targets. The protective effect of miR-133b on cardiomyocyte apoptosis and myocardial fibrosis may be mediated by the inhibition of PTBP1 and TAGLN2 expression [141].

### 4.6. miR-140

miR-140 is one of the miRNAs that play a variety of roles in human diseases [142]. miR-140-5p has been shown to control the inflammatory signaling cascade, such as the toll-like receptor 4 (TLR4)/NF-κB signaling pathway [143]. The upregulated miR-140 also plays an important role in DIC. However, it concurrently aggravates cardiotoxicity by increasing oxidative stress and destroying mitochondria.

Zhao et al. [144] randomly divided 20 rats and 20 mice into a DOX treatment group (15 mg/kg/d, for 8 d) and a control group (0.9% normal saline). In addition, the levels of oxidative stress were detected after H9c2 cells were treated with different doses of DOX (5 μmol/L, for 24 h). The in vivo experiment showed obvious cardiomyocyte injury in the rats treated with DOX, alongside an unclear myocardial tissue texture, pyknosis, and plasmolysis, while the ST-segment change was the most obvious ECG abnormality in the DOX treatment group. Analysis of serum and heart tissue samples revealed that the serum levels of CK and LDH in rats and mice in the DOX group were significantly higher than those in the control group, and there was significant upregulation of miR-140-5p expression. In all DOX-treated H9c2 models, the expression level of miR-140-5p was also significantly increased, and it directly targeted Nrf2 and SIRT2, and increased myocardial oxidative damage. Mitofusin 1 (Mfn1) acts on mitochondria and inhibits mitochondrial fission and apoptosis in cardiomyocytes [145], which is regulated by miR-140; thus, it is a target of miR-140 in the cardiomyocyte apoptosis cascade [146]. Li et al. [147] isolated neonatal rat cardiomyocytes and cardiac fibroblasts from Wistar rats and treated them with DOX (1 μmol/L) for 0–15 h, and miR-140 expression levels were measured after treatment. The expression of miR-140 in DIC was upregulated, which inhibited the Mfn1 expression and aggravated mitochondrial damage.

### 4.7. miR-208

miR-208 is one of the heart-specific miRNAs that play an important role in maintaining contractility by regulating the expression of MHC [45]. It is closely related to the development of heart diseases such as myocardial hypertrophy, myocardial fibrosis, MI, arrhythmia, and HF [148]. The role of miR-208 in DIC is also actively explored; however, there are still more controversial results in the current study.

Novak et al. [120] intraperitoneally injected DOX (5 mg/kg) and L-DOX (5 mg/kg) into rats, then, sacrificed the rats 24h later to obtain LA, LV, RA, and RV. The results showed that both DOX and L-DOX reduced miR-208a in the LV by 38.87% and 23.57% (*p* = 0.028), respectively, meaning that administering DOX produced more damage to miR-208a levels than L-DOX. In contrast, Vacchi-Suzzi et al. [149] used DOX (1, 2, and 3 mg/kg/week) for 2, 4, and 6 weeks, and discovered that miR-208b was slightly upregulated in the hearts of most rats after only 2 weeks of DOX treatment. Furthermore, after 4 weeks of DOX treatment, the level of miR-208b was significantly increased and a dose-dependent expression of miR-208b was observed, while the results at week 6 were inconclusive as only 2 rats survived in the high-dosage group. However, another study [109] showed no increase in plasma miR-208 levels after DOX treatment, which might be related to the absence of cardiotoxicity in the experimental mice. Therefore, research is still needed to further define the relationship between miR-208 and DIC.

### 4.8. miR-499

miR-499 is involved in the coding of myosin and is expressed at high levels in the heart [150]. Similar in function to miR-208, miR-499 can downregulate the expression of β-MHC and enhance the heart’s tolerance to oxidative stress [151]. However, the role of the miR-499 family in tumors is controversial, and studies have shown that miR-499 may be a tumor suppressor gene.

Cheung et al. [110] treated hPSC ventricular cardiomyocytes with DOX (1 μmol/L) for 48 h, which resulted in a dose and time-dependent upregulation of miR-499. Wan et al. [152] divided the experiment into two parts, in vivo and in vitro; for the in vivo experiments, mice were divided into DOX group (15 mg/kg, biw) and control group (normal saline, biw), and the results indicated the mice developed cardiac hypertrophy, and the expression of cardiac hypertrophy markers ANP and β-MHC was upregulated. The level of miR-499-5p in the heart of DOX-treated mice was significantly downregulated (*p* < 0.05). However, the above process was reversed after transfection of miR-499-5p mimic. In vitro studies showed that miR-499-5p began to be downregulated (*p* < 0.05) after DOX (2 μM/L) treatment of H9c2 for 1 h; miR-499-5p mimic bound to the 3’-UTR of p21, significantly downregulating its expression, and weakening mitochondrial fission and DOX cardiotoxicity. At the same time, the study showed that miR-499-5p showed low expression in a variety of tumor cell lines including gastric, lung, colon, and liver cancers compared with cardiomyocytes. miR-499-5p may be a specific biomarker for DIC, but further validation is still needed.

Since miRNAs are stable, widely distributed, and have high sensitivity in the human body, they can be employed as potential diagnostic biomarkers. Several experimental and clinical studies have confirmed the expression of miRNAs in DIC cell models, animal models, or DIC patients, either directly or indirectly. Among them, the expressions of miR-1, let-7, miR-21, miR-34, miR-133, miR-140, miR-208, miR-499, etc., have been studied in relation to DIC, alongside other kinds of miRNAs that are also involved (Table 1).

## 5. miRNA as a Therapeutic Target of DIC

As a therapeutic target of diseases, miRNAs also belong to a relatively new research field. According to the present research situation, the development of diagnosis and treatment methods based on miRNAs is receiving extensive attention [161]. Indeed, miRNAs have certain advantages in the diagnosis and treatment of cardiac toxicity resulting from chemotherapy. For example, as potential biomarkers, miRNAs can be used as the index to evaluate the risk and efficacy of DIC. However, in contrast, the imbalance of miRNA expression plays a key role in the occurrence and development of DIC. When miRNAs itself is the key link that causes DIC, drugs can be developed for its upstream or downstream targets or pathways, and the miRNA expression can be upregulated or downregulated, thus antagonizing DIC. Currently, as a therapeutic target of DIC, miRNAs are more concentrated on antagonizing oxidative stress and reducing apoptosis [162,163].

### 5.1. Antagonistic Antioxidant Stress and Apoptosis

The role of miRNA as a therapeutic target in oxidative stress has been confirmed in multiple studies. Further studies have focused on miRNA antagonists, drugs (Western medicine and Chinese herbal medicine extracts), and lncRNA, which have been demonstrated to up- or downregulate miRNA expression and affect their downstream targets to produce corresponding biological effects, thus further playing a role in protecting the heart and reducing DIC.

miR-1 is abundant in heart tissues; its expression is also closely related to DOX-induced oxidative stress. Li et al. [164] treated H9c2 cells with paeoniflorin (PEF) (100 mol/L) for 2h before exposing them to DOX (5 mol/L) for 24h. Studies have shown that under the action of PET, miR-1 was downregulated, Bcl-2 expression was upregulated, and ROS production was inhibited, which subsequently reduced the occurrence of oxidative stress in cardiomyocytes. Tanshinone IIA is the active ingredient from salvia miltiorha and is used to treat CVD [165] because it has been confirmed to have antioxidant and anti-inflammatory activities in disease model animals [166], in addition to clinical treatment value. Song et al. [167] have shown that a low dose of tanshinone IIA (5 µmol/L) could upregulate the expression of miR-133, while a high concentration of endogenous miR-133 expression was almost completely restored to a level comparable to the control group. It was demonstrated that tanshinone IIA could reduce apoptosis by upregulating miR-133 and inhibiting caspase-9. Irigenin (IR) can reduce DOX-induced fibrosis, cardiac dysfunction, and injury by reducing apoptosis, oxidative stress, and inflammation in cardiac tissue samples [168]. Guo et al. [169] found that DOX caused a significant decrease of miR-425 in HL-1 cells, while IR significantly upregulated the expression of miR-425. Receptor-interacting protein kinase 1 (RIPK1) has been found to be a direct target of miR-425, and DOX induces RIPK1 overexpression both in vitro and in vivo. After IR treatment, RIPK1 overexpression was significantly reduced, which reduced the apoptosis of HL-1 cells and ROS production. Dioscin, which is isolated from herbs, has been shown to strongly protect against organ injuries and to modulate oxidative stress through the regulation of multiple signaling pathways [170,171,172]. According to a study by Zhao et al. [173], DOX (5 μmol/L) treatment significantly increased the expression level of miR-140-5p in H9c2 cells. Likewise, compared with the control group, the intracellular ROS level was increased in H9c2 cells treated with DOX. However, compared with the DOX group, dioscin (50, 100, and 200 ng/mL) inhibited the expression of miR-140-5p, which upregulated the expression of Nrf2 and SIRT2, and alleviated the DOX-induced myocardial oxidative stress. As the only drug approved by the Food and Drug Administration (FDA) for the treatment of DIC, the mechanism of action of dexrazoxane (DEX) is not completely clear. miR-17-5p is associated with oxidative stress [174,175] and plays a cytoprotective role in the hypoxia response by regulating apoptosis. Yu et al. [176] studied the relationship between DEX and miR-17-5p by randomly dividing mice into a control group, DOX treatment group, DOX plus DEX treatment group, and DEX treatment group, and transfected with the miR-17-5p inhibitor. Studies have shown that miR-17-5p significantly reduced the expression of phosphatase and angiotensin homolog in cardiomyocytes exposed to DOX, while DEX reduced cardiomyocyte apoptosis by upregulating miR-17-5p to antagonize oxidative stress.

Bcl-2, an anti-apoptotic protein, significantly inhibited apoptosis and reduced the generation of ROS [177]; SIRT1 is a potential node for regulating DOX-induced cardiotoxicity and can prevent DOX-induced cardiac dysfunction through the oxidative stress pathway [178]. Xu et al. [179] used miR-22 antagonists to significantly reduce the level of miR-22 and upregulate SIRT1 expression, which culminated in the reduction of DOX-induced oxidative stress damage to cardiomyocytes. Li et al. [180] recorded that the use of DOX treatment for 24 h increased the apoptosis of H9c2 cells and the increased expression of miR-25. Western blot results showed that DOX-induced oxidative stress increased Bax levels and decreased Bcl-2 levels, which led to increased ROS production. In the miR-25 inhibitor group, the inhibition of miR-25 resulted in a downregulation of Bax expression and an increase in Bcl-2 levels, which inhibited the production of ROS. Piegari et al. [181] showed that miR-34a is upregulated in DIC models. Following administration of the antagonist, the expression of miR-34a decreased, Bcl-2 and SIRT1 were upregulated, and apoptosis was inhibited. Piegari et al. [182] exposed rats’ CPCs, H9c2 cells, fibroblasts, and aortic endothelial cells to DOX (0.5 μmol/L) for 48h: in all cells, miR-34a was significantly upregulated after DOX treatment. After using the antagonist, the level of miR-34a in CPCs was significantly downregulated, while the Bcl-2 and SIRT1 levels were similarly downregulated, and the p53-acetylated form was significantly reduced. One of the confirmed targets of miR-208a is myocardial transcription factor 4 (GATA4), which increased after DOX treatment and reduced cardiomyocyte apoptosis, and improved cardiac function [183]. Tony et al. [184] showed that after administering the antagonist, miR-208a was downregulated, and GATA4 and Bcl-2 were upregulated, which reduced oxidative stress and myocardial apoptosis. Cell-based therapies have been reported to regulate cell senescence [185]. Xia et al. [186] evaluated the effect of mesenchymal stem cells (MSCs) on DOX-treated H9c2 cells to explore the anti-aging mechanism of MSCs involving the role of MSCs in the miR-34a-SIRT1 axis. In the presence of DOX (0.5 μmol), H9c2 cells are in a state of senescence, characterized by low proliferation, poor viability, and significantly increased expressions of p53 and p16. In contrast, H9c2 cells showed increased proliferation and viability when co-cultured with MSCs in the presence of DOX, and the expression of miR-34a was downregulated, indicating that MSCs could increase SIRT1 expression by inhibiting miR-34a. Fu et al. [187] showed that miR-200a-3p regulates the proliferation and apoptosis of cardiomyocytes by targeting the SIRT1/NF-κB signaling pathway. Inhibition of miR-200a-3p upregulates DOX damage to cardiomyocytes. Another study also showed that inhibition of miR-200a aggravates DOX-induced myocardial apoptosis [188].

The AKT kinase is a member of the AGC kinases that play key and distinct roles in the cardiovascular system [189]. Some studies have demonstrated that miRNA-mediated gene regulation and the AKT pathway are interconnected to form an AKT-miRNA regulatory network [190], which acts on a variety of cellular responses. Li et al. [191] established a DOX-induced cardiotoxicity mice model, where a group of mice were injected with the miR-143 antagonist, and H2c9 cardiomyocytes were exposed to DOX (1 μmol/L) to establish an in vitro model. The in vivo results were consistent with those in vitro and showed that under the action of the miR-143 antagonist, the DOX-induced downregulation of superoxide dismutase (SOD) activity and upregulation of NOX activity were both weakened, reduced lipid peroxidation in the myocardium, and restored glutathione (GSH) levels. In addition, the use of miR-143 antagonists significantly activated AKT, while reducing oxidative stress. Zhang et al. [192] established a mice cardiotoxicity model, and a single subcutaneous injection of the miR-375 inhibitor was performed after treatment. In addition, H9c2 cardiomyocytes and adult mice cardiomyocytes (AMCs) were cultured in vitro and treated with DOX and the antagonists. In vivo experiments showed that the mRNA levels of Bax, which encodes pro-apoptotic molecules, were upregulated, whereas Bcl-2 levels, which encode anti-apoptotic molecules, were downregulated in DOX-treated mouse hearts; further, these effects were both reversed by using miR-375 inhibitors. Consistent with the in vivo data, inhibition of miR-375 significantly reduced the upregulation of 3-NT and MDA in DOX-treated H9c2 cardiomyocytes, and inhibition of miR-375 also prevented the induction of Bax and the inhibition of Bcl-2 expression in the presence of DOX. In addition, miR-375 inhibitors can activate the 3-phosphoinositide-dependent protein kinase 1 (PDK1)/AKT axis and decrease ROS production by reducing the direct binding of miR-375 to the PDK1 gene 3’ UTR. Both in vivo and in vitro experiments have shown that inhibition of miR-375 can re-establish myocardial redox homeostasis and prevent DOX-induced oxidative stress and cardiomyocyte apoptosis.

### 5.2. Reduced Mitochondrial Damage

Mitochondria are the main organelles damaged by DOX cardiomyocytes [193]. DOX can remain in the inner mitochondrial membrane of myocardial cells, and combine with membrane phospholipids to form a complex, which can affect mitochondrial function, destroy mitochondria, and aggravate the DIC [194]. Li et al. [195] showed that after 4 weeks of DOX treatment, BNP levels were significantly increased compared with the control rats (*p <* 0.001). miR-24 is an inhibitor of Junctophilin-2 (JP-2), and high-dose Lingguizhugan decoction (LGZG) significantly downregulated miR-24 expression compared with DOX-treated rats (*p <* 0.001) and upregulated the expression of JP-2 and antagonized DOX-induced microstructural remodeling of T-tubule-sarcoplasmic reticulum (TT-SR). Furthermore, LGZG reduced the size of swollen mitochondria, inhibited the number of mitochondrial fragments, and alleviated morphological abnormalities and mitochondrial damage in the myocardium. Du et al. [196] established the DIC model in primary neonatal rat ventricular myocytes (NRVMS). The results showed that the significantly increased expression of miR-23a was related to DOX concentration. By directly targeting peroxisome proliferator-activated receptor gamma coactivator-1α (PGC-1α) and dynamin-related protein-1 (Drp1) to inhibit miR-23a, the cardiomyocyte injury was reduced, and the mitochondrial-dependent apoptosis was inhibited. 

### 5.3. Involved in DNA Methylation

The main function of DNA methyltransferase 1 (DNMT1) is to restore methylation of newly synthesized daughter DNA strands, which play a key role in the development of smooth muscle cells [197], and its dysregulation is closely related to CVD [198]. Deng et al. established a DOX-induced model of heart failure in rats. Echocardiogram results showed that DOX-treated rats had decreased LVEF and LVFS, LVEDD and LVESD increased, DNMT1 levels increased, and miR-152-3p levels decreased. Methylation-specific PCR (MSP) results showed that DNMT1 inhibited the expression of miR-152-3p by increasing DNA methylation in the miR-152-3p promoter region. Transfection of miR-152-3p mimics significantly reduced the luciferase activity of E26 transformation specific-1 (ETS1)-3’UTR-wt, while ETS1 regulates Ras homolog gene family member H (RhoH) in myoblasts, and DNMT1 may inhibit miR-152-3p expression by promoting methylation inhibition of miR-152-3p and enhancing the expression of ETS1, thereby inducing RHOH transcription activation and inhibiting mitochondrial autophagy, ultimately promoting the development of heart failure [199].

### 5.4. miRNAs Are Transported into the Heart by Exosomes to Interfere with DIC

Therapeutic transmission of protective miRNA is a relatively new treatment, which may block cardiotoxicity [200] Indeed, Sun et al. [201] intraperitoneally injected DOX into mice for 4 weeks. One day prior to DOX treatment, the mice were treated with miR-21-coated exosomes, and the exosomes were tracked using cell-miR-54. The results showed that the cell-miR-54 abundance in the heart was increased approximately 15-fold, indicating that miR-21 entered the heart. It was confirmed that the exosome miR-21 transmission strategy could significantly improve the prevention of DOX-induced myocardial toxicity in the heart. However, there is limited research in this field; thus, we can focus on the exosome-based treatment of DIC in future studies.

Because miRNAs have the characteristics of multi-targets, they are the regulatory factors of various cascade reactions. Therefore, targeting the miRNAs related to DIC is a potential therapeutic means, which can promote or inhibit the occurrence of DIC through various targets and pathways (Table 2).

## 6. Discussion and Prospect

CVD and cancer are the two leading causes of human mortality, accounting for at least 70% of the medical causes of global mortality [208]. With the development and progress of modern medicine, people’s concept of cancer treatment has changed from simply improving survival rate to comprehensively improving the quality of life of patients. Cancer and cardiovascular disease have many common risk factors, and the two often appear together. Many anti-cancer treatment options have potential cardiovascular toxicity, affecting patient prognosis; on the other hand, cardiovascular health concerns can also affect cancer treatment, thus determining how to manage the cardiovascular health of cancer patients became a thorny problem faced by oncologists and cardiologists, and cardio-oncology was born [209]. Since the 1970s, DIC has been continuously reported [210,211]; it is thought to be more dangerous than DOX-induced myelosuppression, digestive tract and renal toxicity, etc., because early DIC is more insidious, and people are often unaware of the cardiotoxicity caused by DOX and lack of monitoring of cardiac function. Studies have found that one of the biggest determinants of the development of heart failure caused by DOX is its cumulative dose. When the cumulative dose of DOX reached 400 mg/m^2^, the incidence of cardiac insufficiency was 3–5%; when the cumulative dose reached 550 mg/m^2^, the incidence of cardiac insufficiency increased to 7–26%; When the cumulative dose reaches 700 mg/m^2^, the incidence of cardiac insufficiency is as high as 18–48% [212,213]. The effect of DOX on cells is non-selective, so cardiotoxicity is inevitable; therefore, early monitoring and intervention are essential for the occurrence, development, and prognosis of DIC. Cardiac biomarkers offer a potential solution to this problem, as they may help in the early identification of subclinical cardiovascular toxicity.

About 98% of the sequence of the DNA of the human genome is transcribed as ncRNAs [214], which were considered “junk RNA” in the early days of discovery because they normally do not encode proteins [215]. With the deepening of research, the physiological and pathological functions of ncRNAs have been gradually discovered. ncRNAs are widely found in a wide range of organisms, from bacteria to mammals, and perform a variety of regulatory functions, such as inhibiting the translation, degradation, and cleavage of mRNAs [216], chromatin remodeling [217,218], and cell cycle regulation [219]. miRNAs have been one of the hot spots in the field of RNA research in recent years. Mature miRNAs mainly negatively regulate the post-transcriptional level of genes, by causing degradation of target mRNA or interruption of the translation process. At the same time, miRNAs have high stability and easy detection, and miRNA-based therapies, whether they are regarded as potential biomarkers, or restore or inhibit the expression and activity of miRNAs to produce corresponding functions, have great clinical application prospects. In recent years, more and more studies have also focused on the dysregulation of miRNAs in cardiotoxicity after antitumor therapy and the pathological changes brought about. Based on the existing research, this article summarizes the role of miRNAs in the physiological and pathological processes of hearts and tumors, and reviews the latest research progress related to miRNA and DIC. miR-1, miR-133, miR-208, and miR-499 are heart-specific miRNAs, and let-7, miR-21, miR-34, etc., play an important role in the physiological and pathological processes of the heart. A number of studies have shown that these miRNAs are upregulated in DIC animal and human cell models, while in patients using DOX chemotherapy, clinical studies have shown that miR-1 is associated with early DIC and is a potential biomarker of early DIC. These miRNAs may be involved in the occurrence of DIC, although the mechanism is not clear. In recent years, miRNA-related DIC therapies have also been continuously explored, such as inhibitors of miRNAs; traditional Chinese medicine monomer components, etc., can effectively target related miRNAs to inhibit their expression and produce corresponding biological functions, and new therapeutic methods such as therapeutic miRNA delivery are also being explored. These studies illustrate the great potential of miRNAs in the diagnosis and treatment of DIC.

However, there are still some limitations in this field. First of all, miRNAs have the characteristics of multiple targets, with an average of 200 targets for each miRNA [220], and widely participate in various biological processes through a plurality of targets. Many studies have not compared the expression levels of miRNAs in healthy or cardiovascular and tumor populations, so simply measuring the expression of miRNAs after DOX treatment is not completely accurate. Furthermore, the dysregulation of the expression of miRNAs in much DIC cannot be confirmed as systemic changes after treatment with DOX, or because of elevated tumor-induced miRNAs; for example, although many studies have reported miR-34 changes in DIC animal or cell models, miR-34 is also dysregulated in many tumor patients and cancer stem cells. Although some miRNAs belong to the specific miRNAs of the heart, such as miR-1/133, there are disorders in a variety of heart-related diseases, including DIC, but the exploration of these miRNAs and the pathogenesis of DIC has not been sufficiently in depth, which limits the application of these miRNAs. In addition, in the diagnosis of DIC, there is a lack of simple and accurate diagnostic methods; even cTn cannot fully reflect cardiotoxicity, because cTn is a paraclinical marker released by the cardiovascular system, which makes many miRNAs have no benchmark to refer to to a certain extent in the diagnosis of DIC. The prognosis of miRNAs also lacks a large number of prospective studies to support the prognosis of DIC, and it is impossible to establish the relationship between the upregulation or downregulation of miRNAs and early DIC or late DIC. There are also some problems in the design of some experiments; some studies only refer to heart samples for study samples and do not indicate whether the whole heart, part of the heart, or only specific heart tissues such as the atria or ventricles were used for the analysis. Finally, the expression of miRNAs in DIC and its expression time need to be further explored; based on existing research, only the expression of these miRNAs in DIC animal models or DIC patients can be judged, but the specific expression time cannot be judged.

In summary, the role of miRNAs in DIC is of great significance, since it is not only a regulation of multiple pathways and targets that mediate DIC, but also a potential biomarker and therapeutic target for DIC. Based on the existing studies showing that miRNAs have the potential to be used as DIC biomarkers and therapeutic targets, the mechanism of miRNAs participating in DIC should be strengthened in future research to further reveal their mechanism of action in DIC, which will be very helpful for realizing the clinical application of miRNAs.

## Figures and Tables

**Figure 1 biomolecules-13-00568-f001:**
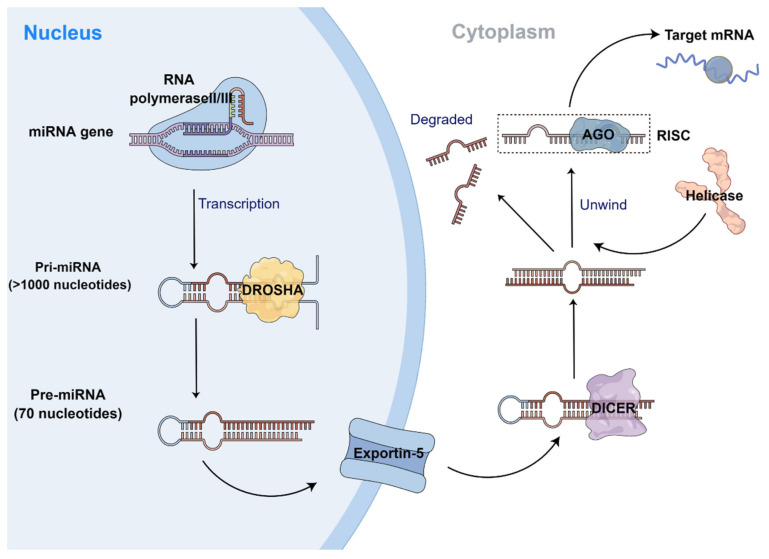
Schematic diagram of synthesis and processing of miRNAs. The gene of the miRNA is transcribed into a pri-miRNA under the mediation of RNA polymerase II/III and then cut into the pri-miRNA by DROSHA; subsequently, the pre-miRNA is transported by exportin-5 into the cytoplasm and cleaved by the endonuclease DICER; then, under the action of helicase, two single strands are formed, one of which forms RISC with AGO, and the other is degraded.

**Figure 2 biomolecules-13-00568-f002:**
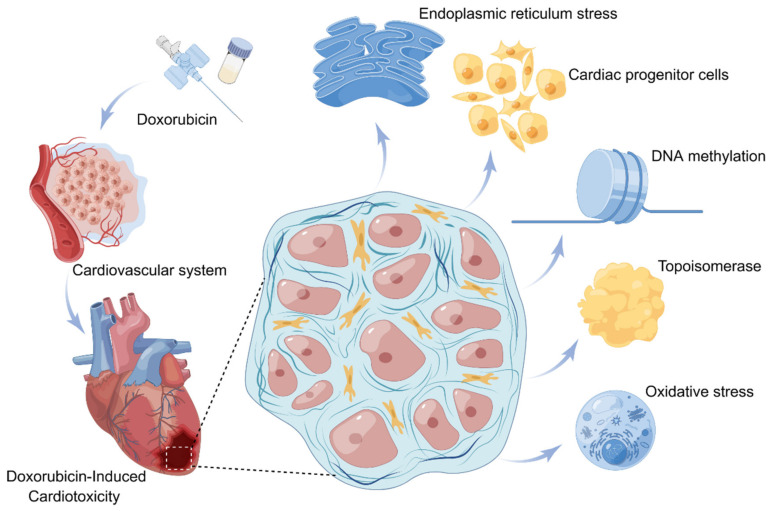
Mechanism diagram of DIC. Five potential mechanisms by which DOX acts on the cardiovascular system and produces cardiotoxicity: oxidative stress; the role of topoisomerase in DIC; DNA methylation; cardiac progenitor cells; endoplasmic reticulum stress.

**Figure 3 biomolecules-13-00568-f003:**
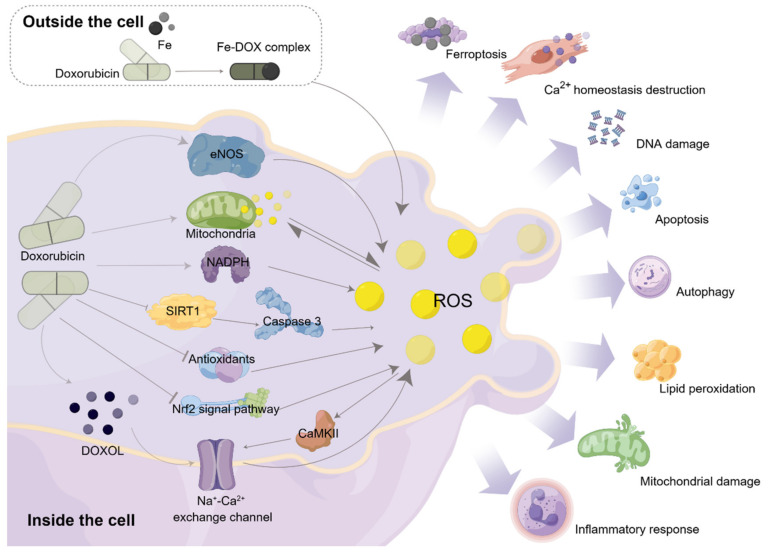
Schematic diagram of the occurrence and development of oxidative stress in DIC. DOX enters the body and produces ROS mainly through the following ways, mediating oxidative stress; DOX combines with eNOS to induce DOX, generating ROS; DOX binds to the inner mitochondrial membrane to form complexes, alters cardiolipin function, and produces ROS; DOX forms Fe-DOX complexes with Fe, which contribute to ROS production; under the action of the NOX enzyme, DOX reacts with NADPH to generate ROS; DOXOL, a metabolite of DOX, inhibits the sodium-calcium exchange channel, causing an imbalance in the level of Ca^2+^ in the cell, and produces ROS, which inhibits CaMKII and aggravates the imbalance of Ca^2+^ level; excessive consumption of antioxidants; downregulation of Nrf2 pathway expression; DOX treatment significantly inhibited SIRT1 deacetylase activity and protein levels, and promoted deubiquitination of p53 and activation of caspase-3. The above seven ways are involved in DOX-induced oxidative stress, individually or synergistically.

**Table 1 biomolecules-13-00568-t001:** Expression of miRNAs as biomarkers in DIC.

miRNA	Source	Species	Change	Cardiotoxicity Assessment	Compared with Traditional Biomarkers	References
miR-1	Mice	Heart tissue/serum	Up	Lactate Dehydrogenase (LDH)	Overmatch	[108]
	Rats	Serum	Up	cTnI, cTnT	Overmatch	[109]
	Cell strain	hPSC-CM	Up	N/A	N/A	[110]
	Human	Plasma	Up	cTnT	Overmatch	[111]
	Human	Plasma	Up	cTnI, LVEF	Overmatch	[112]
	Human	Plasma	Up	GLS, cTnT	N/A	[110]
	Human	Plasma	Unclear	LVEF	N/A	[113]
let-7	Rats	Plasma	Down	Heartrate, Pulse pressure, cTnT	N/A	[118]
	Cell	Cardiomyocyte	Down	N/A	N/A	[118]
	Cell strain	H9c2 cell	Down	N/A	N/A	[119]
	Rats	Heart tissue	Down	N/A	N/A	[120]
	Human	Plasma	Down	LVEF	N/A	[113]
miR-21	Mice	Heart tissue/serum	Up	LDH	Overmatch	[108]
	Mice	Cardiomyocyte	Up	LVEF, LDH	N/A	[126]
miR-29b	Rat	Heart tissue	Down	LVEF	N/A	[153]
miR-34a-5p	Mice	Plasma	Up	cTnT	N/A	[130]
	Mice	Heart tissue/plasma	Up	LVEF	N/A	[131]
	Cell strain	HL-1	Up	N/A	N/A	[131]
	Mice	Plasma	Up	N/A	N/A	[132]
	Mice	Plasma	Up	N/A	N/A	[132]
	Cell strain	H9c2 cell	Up	N/A	N/A	[136]
miR-34b-3p	Cell strain	HL-1	Up	N/A	N/A	[131]
miR-34c-5p	Cell strain	HL-1	Up	N/A	N/A	[131]
miR-34a	Mice	Heart tissue/plasma	Up	cTnT	Overmatch	[134]
	Human	Plasma	Up	cTn, LVEF	Overmatch	[137]
	Cell strain	hES-CM	Up	N/A	N/A	[133]
miR-34b	Cell strain	hES-CM	Up	N/A	N/A	[133]
	Cell strain	HL-I	Up	N/A	N/A	[135]
	Cell strain	HL-I	Up	N/A	N/A	[135]
miR-130a	Human	Plasma	Up	LVEF, cTnI, NT-pro BNP	N/A	[154]
miR-133b	Cell strain	HL-1	Down	N/A	N/A	[141]
miR-140-5p	Rat/mice	Heart tissue/plasma	Up	N/A	N/A	[144]
	Cell strain	H9c2	Up	N/A	N/A	[144]
miR-140	Rat	Cardiomyocyte	Up	N/A	N/A	[147]
miR-181c	Mice	Heart tissue	Down	N/A	N/A	[155]
miR-182-5p	Cell strain	hiPSC-CM	Down	LDH	N/A	[156]
miR-187-3p	Cell strain	hiPSC-CM	Down	LDH	N/A	[156]
miR-194-5p	Cell strain	H9c2	Up	N/A	N/A	[157]
miR-199a-3p	Cell strain	hPSC-CM	Down	N/A	N/A	[158]
miR-204	Cell strain	H9c2	Down	cTnI, CK-MB, LDH	N/A	[159]
miR-208a	Rat	Heart tissue	Down	N/A	N/A	[120]
miR-208b	Rat	Heart tissue	Up	N/A	N/A	[149]
miR-208	Mice	Plasma	Up	cTnI, cTnT	N/A	[109]
miR-499	Cell strain	hPSC-CM	Down	N/A	N/A	[110]
	Mice	Cardiomyocyte	Down	LVEF	N/A	[152]
	Cell strain	H9c2 cell	Down	N/A	N/A	[152]
miR-532-3p	Rat/mice	Cardiomyocyte	Up	N/A	N/A	[160]

**Table 2 biomolecules-13-00568-t002:** As the therapeutic target of DIC, the regulatory role of miRNA in DIC.

miRNA	Source	Sample	Intervention Drug	Target/Pathway	Function	References
miR-1	Cell strain	H9c2 cell	Paeoniflorin	Bcl-2	Oxidative stress, apoptosis	[164]
miR-17-5p	Rat	Cardiomyocyte	Antagonist	PTEN	Oxidative stress, apoptosis	[176]
miR-21	Mice	Heart tissue	Exosome-mediated miR-21	N/A	Apoptosis	[201]
miR-22	Mice	Cardiomyocyte	Antagonist	SIRT1	Oxidative stress, apoptosis	[179]
miR-23a	NRVMs	Cardiomyocyte	Antagonist	PGC-1α/p-Drp1	Protect mitochondria	[196]
miR-24	Rat	Cardiomyocyte	LGZG	JP-2	Protect mitochondria	[195]
miR-25	Cell strain	H9c2 cell	Antagonist	Bax, Bcl-2, PTEN	Oxidative stress, apoptosis	[180]
miR-34a	Rat	Cardiomyocyte	Antagonist	Bcl-2, SIRT1	Oxidative stress, apoptosis	[181]
	Rat	CPCs	Antagonist	Bcl-2, SIRT1, p53	Apoptosis	[182]
	Cell strain	H2c9 cell	MSCs	p53, p16, SIRT1	Apoptosis	[186]
miR-96	Rat	Cardiomyocyte	Antagonist	NF-κB	Apoptosis	[202]
miR-128-3p	Mice	Cardiomyocyte	Antagonist	PPAR-γ, Bax	Oxidative stress, apoptosis	[203]
miR-133	Cell strain	H2c9 cell	Tanshinone IIA	caspase-9	Apoptosis	[167]
miR-143	mice	Cardiomyocyte	Antagonist	BCL-2, BAX, caspase-3	Oxidative stress, apoptosis	[191]
	Cell strain	H2c9 cell	Antagonist	BCL-2, BAX, caspase-3, PKB, AKT	Oxidative stress	[191]
miR-140-5p	Cell strain	H2c9 cell	Dioscin	Nrf2, Sirt2	Oxidative stress, apoptosis	[173]
miR-152-3p	Rat	Cardiomyocyte	DNMT1	ETS1, RhoH	DNA methylation	
miR-199	Mice	Cardiomyocyte	Antagonist	Bax, Bcl-2, TAF9b	Oxidative, autophagy	[204]
miR-200a	Cell strain	H2c9 cell	Antagonist	Keap1, Nrf2	Oxidative stress, apoptosis	[188]
miR-200a-3p	Rat	Cardiomyocyte	Antagonist	SIRT1/NF-κB, PEG3	Apoptosis	[187]
miR-375	Rat	Cardiomyocyte	Antagonist	Bax, Bcl-2	Oxidative stress, apoptosis	[192]
	Cell strain	H2c9 cell, AMCs	Antagonist	PDK1, AKT	Oxidative stress, apoptosis	[192]
miR-208a	Mice	Heart tissue	Antagonist	GATA4, Bcl-2	Oxidative stress, apoptosis	[184]
miR-425	Mice	plasma	Irigenin	Bcl-2, Bax, Caspase-3	Oxidative stress, apoptosis	[169]
	Cell strain	HL-1	Irigenin	Bcl-2, Bax, Caspase-3, PARP	Oxidative stress, apoptosis	[169]
miR-495-3p	Cell strain	H2c9 cell	Irigenin	PKB/AKT	Oxidative stress, apoptosis	[205]
miR-875-3p	Cell strain	H2c9 cell	lncRNA HOXB-AS3	N/A	Apoptosis	[206]
miR-1303	Cell strain	A16	Kinetochore-associated protein 3	TLR4	Apoptosis	[207]

## Data Availability

The research data are not involved in this study.

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
