# Peer review of "MicroRNA in the Diagnosis and Treatment of Doxorubicin-Induced Cardiotoxicity"

_biomolecules, 2023, doi:10.3390/biom13030568_

Round 1
Reviewer 1 Report
I have reviewed the manuscript entitled 'MicroRNA in the diagnosis and treatment of doxorubicin-induced cardiotoxicity'.
The manuscript is well-written and presented however several issues should be checked before further evaluation.
First, please check spelling and typo errors.
Second, there should be a scoring system to identify patients before the development of heart failure following doxorubicin or similar drugs. A new index has been reported to have prognostic value in patients with diastolic heart failure. Please add a short section citing 'A simple formula to predict echocardiographic diastolic dysfunction-electrocardiographic diastolic index'.
Author Response
Response to Reviewer 1 Comments
Point 1:First, please check spelling and typo errors.
Response 1: Thanks for your suggestions. The language was edited with the help of Wordvice.
Point 2: Second, there should be a scoring system to identify patients before the development of heart failure following doxorubicin or similar drugs. A new index has been reported to have prognostic value in patients with diastolic heart failure. Please add a short section citing 'A simple formula to predict echocardiographic diastolic dysfunction-electrocardiographic diastolic index'.
Response 2: Thank you for your valuable advice, we have made adjustment based on your suggestion. “A simple formula to predict echocardiographic diastolic dysfunction-electrocardiographic diastolic index” was cited in “3.2. Diagnosis and treatment of DIC” (page 8, [100]).
Reviewer 2 Report
Dear Editor
The review submitted by Klang et al. is a collection of information on miRNA changes after treatment with doxorubicin.
The review is well-written and attempts to provide readers with information about the diagnostic and therapeutic potential of CVD-associated miRNA.
I have only a few comments that I hope will help improve the review and address the questions and concerns of Biomolecules readers.
These miRNAs should be identified and measured in cancer patients. It is not clear how the changes in these miRNAs can be distinguished between systematic responses to the drug or specifically related to heart diseases.
Comparison of these miRNAs with troponin is also not directly accurate because troponin is a paraclinical marker released by the cardiovascular system.
Increased levels of certain miRNAs after chemical treatment are to be expected, but whether these changes affect the cardiovascular system is not clear from the current version of the review.
The authors need to discuss this question further.
Following the same paradigm, the authors limit their review to the detection of miRNAs and do not discuss the potential mechanisms leading to an increase in miRNAs associated with CVD. With the exception of miRNA-let7 and miRNA-140, we read nothing about target genes or proteins involved in CVD.
Since miRNAs have been discussed in the context of cancer therapy, it would be good if the authors provided information on such changes after combined radio- and chemotherapy.
Both figures need figure legends with brief descriptions of what is seen and implied in the figure.
Please include information about the consequences of decreased DNA methylation for the development or progression of CVD (page 11).
Please explain how early or how late these selected miRNAs are detectable.
How are they associated with the late or early effect of DOX on the heart?
Is there any information on which cardiac cell types are responsible for the release of miRNAs and which are most responsive to them?
Yours sincerely
Author Response
Response to Reviewer 2 Comments
Point 1: These miRNAs should be identified and measured in cancer patients. It is not clear how the changes in these miRNAs can be distinguished between systematic responses to the drug or specifically related to heart diseases.
Response 1:Thank you for your valuable advice, your suggestion is very good and we paid attention to that. Many studies have ignored the expression of miRNA in patients with cancer, cancer animal models, or cancer cell lines when exploring the expression of miRNA after DOX treatment, which is a limitation of the current studies. We discussed this further in the discussion(page 19).
Point 2: Comparison of these miRNAs with troponin is also not directly accurate because troponin is a paraclinical marker released by the cardiovascular system.
Response 2: Thank you for your valuable advice, we have made adjustments based on your suggestion, and we discussed this further in “3.2. Diagnosis and treatment of DIC” (page 8) and discussion (page 19).
Point 3 :Increased levels of certain miRNAs after chemical treatment are to be expected, but whether these changes affect the cardiovascular system is not clear from the current version of the review.
Response 3:Thank you for your valuable advice, we are also concerned about this, which is indeed a problem in this field of research. We discussed this further in the discussion (page 19).
Point 4: Following the same paradigm, the authors limit their review to the detection of miRNAs and do not discuss the potential mechanisms leading to an increase in miRNAs associated with CVD. With the exception of miRNA-let7 and miRNA-140, we read nothing about target genes or proteins involved in CVD.
Response 4: Thanks for your suggestion. The details of potential mechanisms leading to an increase in miRNA were added in the revised manuscript according to your suggestion. The details of relevant target information for miRNAs can be seen in “4. miRNA as a biomarker of DIC' (pages 8 to 13), and two miRNAs were added, miR-133 (page 11) and miR-499 (page 13).
Point 5: Since miRNAs have been discussed in the context of cancer therapy, it would be good if the authors provided information on such changes after combined radio- and chemotherapy.
Response 5: Our study is based on cardiotoxicity caused by doxorubicin, a chemotherapeutic agent, and if radiotherapy is reintroduced, we worry that too many variables may be more difficult to explain the causes of miRNA changes, so we do not elaborate more on radiotherapy.
Point 6: Both figures need figure legends with brief descriptions of what is seen and implied in the figure.
Response 6: Thank you for your valuable advice, we have made adjustments to the figures based on your suggestion.
Point 7: Please include information about the consequences of decreased DNA methylation for the development or progression of CVD (page 11).
Response 7: Thank you for your valuable advice, we have made adjustments based on your suggestion. We have included information about the relationship between miRNAs and DNA methylation in DIC as described in “5.3. Involved in DNA methylation”(page 17).
Point 8: Please explain how early or how late these selected miRNAs are detectable.
Response 8: Thank you for your valuable advice, the majority of research we cited was only for the detection of miRNA at fixed times (like 24 hours, 1 week, etc), contained limited detection time information, and we have supplemented it in the previous version. The information about the detection time of miRNAs described in the original version has been marked in blue, and the newly added content has been marked in red(pages 8 to 13).
Point 9: How are they associated with the late or early effect of DOX on the heart?
Response 9: Thank you for your valuable advice, It’s a pity that current clinical studies showed that only miR-1 was a potential biomarker of early DIC (acute DIC). The relationship between miRNAs and early DIC and late DIC is considered to be a focus of future research. We mentioned this in the discussion section(pages 19 to 20).
Point 10: Is there any information on which cardiac cell types are responsible for the release of miRNAs and which are most responsive to them?
Response 10: Thank you for your valuable advice, we have added the contents you advised in "2.3. miRNAs in cardiovascular physiology and pathology”(page 3).
Round 2
Reviewer 1 Report
Thank you for the required revisions.
Author Response
Thank you for the review.